# Childhood Adversities and Psychological Health of Adult Children of Parents with Mental Illness in Japan

**DOI:** 10.3390/healthcare11020214

**Published:** 2023-01-10

**Authors:** Masako Kageyama, Taku Sakamoto, Ayuna Kobayashi, Akiko Hirama, Hiroyuki Tamura, Keiko Yokoyama

**Affiliations:** 1Institute of Advanced Co-Creation Studies, Osaka University, 1-7 Yamadaoka, Suita 565-0871, Osaka, Japan; 2KODOMO-PEER Tonoxbuilding, 3-5-1 Hirata, Ichikawa 272-0031, Chiba, Japan; 3Department of Nursing, Faculty of Nursing, Yokohama Soei University, 1 Miho-cho, Yokohama 226-0015, Kanagawa, Japan

**Keywords:** adverse childhood experiences, young carers, emotional care, mental disorders

## Abstract

In this study, we seek to clarify whether the present-day experience of psychological distress among adults whose parents suffered from mental illness is related to their childhood experiences of abuse and neglect and their provision of emotional care for their parents during their school-age years. To this end, a web-based cross-sectional study was conducted. A total of 120 participants over the age of 20 who attended a self-help group responded (50% response rate); of these, 94 had a parent diagnosed with a mental illness, and these participants were included for data analysis purposes. Of the 94 respondents, 65 (69.2%) were highly distressed, as measured by a Kessler (K) 6 measure of ≥5. A logistic regression analysis revealed that the experience of providing emotional care for parents during school-age childhood was significantly related to high levels of distress in adulthood (OR = 3.48; 95% CI 1.21–9.96). For children of parents with mental illnesses, the effects of providing emotional care for parents during childhood may include long-term psychological distress. For this reason, mentally ill parents raising children need visiting community nurses or other professionals to provide emotional care on behalf of their children.

## 1. Introduction

Early life experiences make an important contribution to the health of individuals throughout their course of life. Adverse early life experiences such as child abuse and neglect have been a topic of political and academic interest since the 1960s [1]. Child abuse, neglect, and several other such experiences have been labeled as adverse childhood experiences (ACEs), which are defined as potentially traumatic events that occur in childhood (0–17 years). These include experiences of violence, abuse, or neglect; witnessing violence in the home; or having a family member attempt or die by suicide [2]. The cumulative effects of exposure to potentially traumatic events on the developing brain’s stress response result in impairment in the development of brain structures and functions [3]. Decades of research have identified a robust, dose–response relationship between child abuse, neglect, and other forms of ACEs and an increased risk of physical and mental health problems across the lifespan [4,5,6,7].

One of the earliest and largest investigations into ACEs was the CDC-Kaiser ACE study [2], in which ACEs were categorized into three groups: abuse, neglect, and household challenges [2]. The category of household challenges included violence towards mothers, substance abuse in the household, mental illness in the household, parental separation or divorce, and incarcerated household members [5]. In the present study, among the ACEs, we focus on parental mental illness as a household challenge facing children, as well as abuse and neglect. In Japan, studies of ACEs have been scarce to date; however, there have been reports on the long-term effects of ACEs in adulthood in relation to self-harm ideation [8], inappropriate parenting by ACE-affected adults [9], and the onset of mental illness [10]. Regarding the association of ACEs with the development of mental illness in Japan, ACEs have been strongly associated with the development of mental illness in childhood but only weakly with the development of such illness in adulthood [10]. In Japan, the number of reports of child abuse and neglect and of patients with ACE-related mental illness continues to increase [11]. Because little is known of any specifically Japanese characteristics among the long-term effects of ACEs in adulthood, more research needs to be undertaken in this regard.

In this study, among the household challenges category of ACEs, we focus on parental mental illness. In western countries, a substantial amount of research has been conducted on children of parents with mental illness (COPMIs). One meta-analysis revealed an increased risk of developing a mental illness among the children of parents who suffer from severe mental illness [12]. Parental mental illness does not in itself guarantee psychosocial problems in their children. However, there is strong evidence that parents with mental illness are more likely to abuse, neglect, or maltreat their children [13]. In Japan, between one-third and one-half of child abuse cases where child protection is required involve parents with mental health problems [14]. Parental mental illness is known to be a high-risk factor for child abuse in Japan.

Nevertheless, the long-term effects on COPMIs that persist into adulthood have been little studied in western countries to date [15]. Some qualitative studies have reported long-term problems affecting COPMIs as adults, such as interpersonal problems [16], psychological distress [16], and difficulties in raising their own children [15]. Similarly, in Japan, there has been little research on long-term problems affecting COPMIs as adults, although one recent qualitative study reported long-term effects such as psychological distress, difficulties with emotions, an inability to trust people, and trauma persisting into adulthood [17].

In recent years, the experience of being a young carer has been identified in Japan as a particular ACE among COPMIs. In western countries generally, since the mid–1980s, there has been an increased awareness of the concept of young carers [18]. Young carers are defined as ‘children and young persons under 18 who provide or intend to provide care, assistance, or support to another family member’ [18]. Three meta-syntheses of qualitative studies describing children’s subjective experiences revealed a relationship between children’s experiences of performing housework and providing emotional care for their parents and feelings of distress among the children involved [16,19,20]. In a 2022 survey of Japanese high school students, high school students who cared for a family member with a disability or illness were significantly more distressed than those without such responsibilities [21].

This concept of young carers is especially relevant in the study of COPMIs. According to a national survey of young carers conducted in the United Kingdom, 29% of people who require care and 50% of mothers who require care have mental health problems [22]. Regarding young Japanese carers, the first national survey conducted in 2020 revealed that, among parents being cared for by their children, 14.3% of those cared for by middle- school-aged children (13–15 years) and 17.3% of those cared for by high-school-aged children (16–18 years) suffered from a mental illness [23]. Taking on such a responsibility can have physical, social, educational, and emotional impacts on the lives of children [24]. Among the many roles played by young carers, the provision of emotional care is more common when the person receiving the care has mental health problems [22], and this includes ‘observing the care recipients’ emotional state, providing supervision, or trying to cheer them up when they are depressed, etc.’ [22]. Because changes in the behavior and personality of persons with mental illness can be distressing for close family members [25,26,27,28,29], the provision of emotional care can lead to psychological distress for carers, resulting in a lower quality of life [30]. For these reasons, in this study, we focus on the provision of emotional care by COPMIs and on any psychological distress as a result. We sought to identify those factors most likely to lead to psychological distress and possible means to alleviate their effects to improve the quality of life of carers.

This study aims to clarify whether the present-day experience of psychological distress among adult COPMIs is related to their experiences of being abused and neglected and of having provided emotional care to their parents during their childhood years, adjusted for factors potentially associated with dependent and independent variables, as follows: gender [31], age [32], presence or absence of siblings [32], personal experience of mental illness [33], physical condition [34], separation from parents in their childhood [6], having a parent without illness [22], the treatment status of their parents in their childhood [35,36], and the absence of a parental spouse in their childhood [7].

## 2. Methods

### 2.1. Research Design

The present study involves the analysis of data from the web-based, cross-sectional ‘Survey of Children Raised by Parents with Mental Illness’ [37], previously conducted by ourselves. This survey sought to obtain basic information to assess the support methods of COPMIs of elementary and high-school age. Respondents were asked to report their current life situation and to recall past childhood events.

### 2.2. Procedure

The survey was carried out among participants in the biggest self-help group in Japan (KODOMO-PEER) for the adult children of parents with mental illness. A web-based survey (URL supplied in the email) was emailed in October and November 2019. The REDCap (Research Electronic Data Capture) system was used for the web survey.

### 2.3. Study Samples

The inclusion criteria for this survey were as follows: individuals who had participated in KODOMO-PEER at least once in the past and who were at least 20 years old. The exclusion criterion covered all individuals less than 20 years old. Those eligible to participate in KODOMO-PEER are individuals whose parents have a mental illness that may or may not have been diagnosed by a psychiatrist.

Of the 240 people invited to participate in the survey, 120 responded (response rate 50%). In order to limit our analysis to the experiences of young carers (those who had to care for a parent when aged less than 18 years), two participants were excluded because one reported that their parent’s mental illness began after they had graduated from high school and another did not supply an onset time. In addition, to limit the analysis sample to those whose parents had been diagnosed with a mental illness by a psychiatrist, we excluded nineteen participants whose parents were undiagnosed and five participants whose parents’ diagnosis was unknown. In all, we included 94 respondents whose parents developed a diagnosed mental illness before the respondents themselves reached the age of 18.

### 2.4. Measures

#### 2.4.1. Psychological Distress

We used the Kessler (K) 6, a short screening questionnaire consisting of six items that is used to screen for non-specific psychological distress [33]. The questions were designed to ask about psychological distress over the previous 30 days. They include ‘Did you often feel nervous’? and ‘How often did you feel hopeless’? as well as four other items. The degree of psychological distress is measured on a 5-point Likert scale with the following options: 0 (none of the time), 1 (a little of the time), 2 (some of the time), 3 (most of the time), and 4 (all of the time); possible scores, therefore, range from 0–24. The reliability and validity of the Japanese version of the K6 have previously been evaluated, and the two best cut-off points have been estimated as follows: 4/5, as an optimal lower threshold cut-point indicative of moderate mental distress; and 12/13, as a screening cut-off point indicative of severe mental illness [38]. For this study, we chose the 4/5 cut-off point because we did not screen for mental illness. The Cronbach’s alpha was 0.92. The K6 score was used to categorize respondents into two groups: those with high (K6 ≥ 5) and low (K6 ≤ 4) levels of distress.

#### 2.4.2. Independent Variables

We used three independent variables: exposure to aggressive acts, neglect, and emotional care.

Exposure to aggressive acts included experiences of child abuse. However, the survey did not ask about child abuse directly. Respondents were asked to consider the following two items with respect to their school-age life experience: ‘There were constant fights between the adults at home’ and ‘There was an attack from my parents on me or my siblings’. These were operationally defined as ‘exposure to aggressive acts’. If one or both items were selected, respondents were considered as having experienced exposure to aggressive acts.

In terms of neglect, those who chose one or both of the following items—‘An adequate amount of food was not provided’ or ‘The laundry and cleaning were not well done’—were considered to have experienced ‘neglect’.

Emotional care is the most common care role carried out by COPMIs [37]. Those who selected ‘I provided emotional care, such as being close to my parents’ as indicative of their childhood experience were deemed to fall into the ‘emotional care’ category.

Regarding the three variables, respondents were asked to recall and answer in terms of their childhood experience during their periods at elementary school, junior high school, and high school at the ages of 7–12, 13–15, and 16–18 years, respectively.

#### 2.4.3. Control Variables

The control variables of adult COPMIs included gender (male/female), age (20–29/30–39/40–49/50 or older), having siblings (yes/no), suffering mental illness (suffered/not suffered), physical condition (poor/not poor), and separation from parents (experienced/not experienced). The variables related to the parents of respondents when the latter were of school age included whether one or both parents had a mental illness (one or both), whether the illness was treated (continued/discontinued or never treated), and the presence or absence of a spouse (presence/absence).

### 2.5. Data Analysis

First, the frequency distribution of variables was confirmed. Next, the background characteristics of high- and low-distress groups were compared using χ^2^ tests. Finally, to examine the association between distress and the independent variables, a multiple logistic regression was performed. The high- and low-distress groups, as established by the K6 scores, were identified as the dependent variables. Exposure to aggressive acts, neglect, and emotional care were identified as the independent variables. All others were considered control variables after confirming variance inflation factors (less 2.0) due to possible multicollinearity among the variables. When multiple logistic regression was conducted, three independent variables and control variables were selected by the stepwise method. A post hoc logistic regression analysis was performed with a power of 0.81 using G*Power. All analyses, except for the power analysis, were performed in SAS Version 9.4 (SAS Institute Inc., Cary, NC, USA).

### 2.6. Ethical Approval

We explained in the text the purpose and method of the study, that participation in the study was voluntary, and that participants could withdraw at any time if they felt discomfort in recalling their past experiences while responding. Informed consent was given by means of this written explanation; explicit consent was then obtained using a checkbox. The anonymity and confidentiality of participants were maintained as researchers could not determine the email addresses and answers of individual respondents. The Research Ethics Committee, Faculty of Medicine, Osaka University, approved the study protocol on 29 July 2019 (ID: 19152).

## 3. Results

### 3.1. Respondents’ Demographics

As shown in Table 1, of the 94 respondents, 84.0% were female and 16.0% were male. In terms of age, those in their twenties formed the largest group, comprising 37.2% of the total. Of the remainder, 22.3% were in their thirties, 17.0% were in their forties, and 23.4% were aged fifty or more; 57.4% of respondents had siblings, and 42.6% did not. Concerning their present state of health, 24.5% stated that they suffered from mental illness, and 41.5% reported being in poor physical condition. Forty-one respondents (43.6%) had been separated from their parents for more than one month before graduating from high school; of these, twenty-eight (29.8%) reported that their parents had been admitted to psychiatric hospitals. None of the respondents reported that they had been under the child protection system as children.

With regard to which of their parents had been diagnosed with mental illness by a psychiatrist, 67.0% stated only their mothers and 17.0% stated only their fathers, while 16.0% said both parents had been diagnosed. The parents’ diagnoses included schizophrenia (61.7%), depression (22.3%), anxiety disorder (12.8%), bipolar disorder (14.9%), developmental disorders (8.5%), personality disorders (9.6%), alcoholic addiction (5.3%), and others (14.9%) (duplicated). Seventy (74.5%) respondents estimated that the onset of their parents’ illness was when they were younger than elementary-school age (6 years or younger), while eighteen (19.2%) stated that it began when they were at elementary school (age 7–12) and six (6.4%) recalled that it started when they were in junior high school (age 13–15). Concerning treatment for mental illness, 52.1% of respondents recalled that their affected parents were under continuous medical treatment, while 47.9% recalled either discontinued treatment or the absence of treatment. A total of 36.2% of respondents recalled the absence of a parental spouse due to divorce or other reasons. Control variables are shown in Table 1.

### 3.2. Psychological Distress

The average K6 score of the 94 respondents was 9.38 (SD 6.75). A total of 65 respondents (69.2%) reported K6 scores of over 5 (high-distress group), while 29 (30.9%) reported scores of 4 or less (low-distress group).

### 3.3. Independent Variables

As shown in Table 2, a total of 78.7% of respondents were exposed to aggressive acts when they were of school age, with 70.2% being exposed during their elementary school period, 71.3% when in junior high school, and 53.2% when in high school.

A total of 48.9% of respondents had experienced neglect, with 33.0% recalling neglect during their elementary schooling, 37.2% experiencing it while in junior high school, and 26.6% during high school.

Most respondents (73.4%) recalled providing emotional care for parents during their school-age childhood, with 59.6% providing such care during elementary school, 60.6% during junior high school, and 60.6% during high school.

### 3.4. Comparisons between the High-Distress and Low-Distress Groups

As shown in Table 1, compared to the low-distress group, respondents in the high-distress group were significantly younger (*p* = 0.027) and more likely to be in poor physical condition (*p* < 0.001). There were no other significant differences between respondent groups in terms of their gender, the presence or absence of siblings, their own experiences of mental illness, experiences of separation from parents, whether one or both parents had a mental illness, whether the parental illness was treated or not, or whether there was parental spousal absence.

As shown in Table 2, upon comparing independent variables between the high-distress and low-distress groups, we found that individuals in the high-distress group were significantly more likely than those in the low-distress group to have been exposed to aggressive acts while in elementary school (*p* = 0.033) and throughout their school-age childhood (*p* = 0.008). They were also more likely to have provided emotional care while in elementary school (*p* = 0.016), junior high school (*p* = 0.003), high school (*p* = 0.036), and throughout their school-age period (*p* = 0.008).

### 3.5. Independent Variables and Distress

As a result of multiple logistic regression with three independent variables and control variables selected via the stepwise method, we found significantly more respondents who had provided emotional care (OR = 3.48; 95% CI 1.21–9.96) among the independent variables and significantly more respondents in poor physical condition (OR = 7.15; 95% CI 2.17–23.58) among the control variables in the high-distress group (Table 3). The other independent variables, i.e., exposure to aggressive acts and neglect, were not significantly associated with psychological distress. Similarly, the other control variables, i.e., gender, age, having siblings, suffering mental illness as a COPMI, experiences of separation from parents, having one or both parents with mental illness, parental treatment or lack of treatment, and parental spousal presence, were not significantly associated with psychological distress.

## 4. Discussion

In this study, we sought to clarify whether any present-day psychological distress experienced by adult COPMIs is related to their experiences of being abused and neglected and of having provided emotional care for their parents during their school-age years. The main finding of this study is that the experience of providing emotional care to parents during childhood is significantly related to high levels of distress in adulthood.

### 4.1. Study Respondents’ Demographics and Distress

The parents of most respondents (93.7%) experienced the onset of their mental illness before the respondent had graduated from elementary school. As a result, many respondents grew up experiencing their parents’ illness. In terms of gender, the majority of the parents with mental illness were mothers (83.0%) and most of the respondents were daughters (84.0%). In Japan, primary carers for persons with mental illness are mostly women [39]. A Japanese national survey of junior high and high school students found that girls spend more time caring for other family members and bear a greater burden than boys [23]. In Japan, care roles are not only for adults but also for children, where a gender bias has arisen. It is thought that women who have more experience caring for their parents participate more in self-help groups to share their caregiving experiences.

In our survey, more than half of the respondents had parents who suffered from schizophrenia (61.7%), and 47.9% of the respondents reported that their parents’ mental illness was untreated during the respondents’ childhood. In general, only 21.9% of individuals with a mental illness seek treatment [40], and over three-quarters of children (78.5%) have parents who suffer from a mental illness but do not receive mental health care [41]; these findings are similar to those of the present study. The emotional care for untreated parents provided by COPMIs can be a heavy burden, and support for such caregivers is needed. However, none of our respondents reported being taken up by the child protection system, even when they had been neglected or exposed to mildly aggressive acts. In such cases, it can be difficult to identify children who need support because they are not being severely—and, thus, obviously—abused.

In the present study, the average K6 score among respondents was 9.38 (SD 6.75) and 65 (69.2%) respondents produced a K6 score of over 5 (high-distress group), which is higher than in the general population (3.6, SD 3.9) [42] or among carers for the elderly (4.29, SD 4.46) [43] or among general high school students (6.51, SD 5.87) [21]. In this study, 24.5% of respondents themselves suffered from a mental illness, which is higher than in the general population (7.6% in 12-month prevalence) [40], and there was no significant relationship between their diagnosis and their level of distress. One meta-analysis found that children of parents with severe mental illness had a 32% probability of developing severe mental illness by adulthood (age > 20) [12]. Regarding the association of parental mental illness with the development of mental illness in their children, in Japan, there appears to be a strong association with the development of such illness during childhood but not in adulthood [10]. A high prevalence of mental illnesses has also been reported among COPMIs under the age of 18, with a significantly higher risk of socioeconomic adversity as a background [44]. It may be that the parent’s mental illness does not itself lead to the development of mental illness in their child. Rather, it is the deterioration of the domestic environment accompanying the onset of the parent’s mental illness that negatively impacts the child’s mental health.

### 4.2. Relationship between Aggressive Acts, Neglect, and Provision of Emotional Care and Levels of Distress

In our high-distress group, we found significantly more respondents who had provided emotional care. There was no significant relationship between childhood exposure to aggressive acts or neglect and present-day levels of distress. Studies on ACEs have identified significant relationships between child abuse or neglect and an increased risk of psychological health problems throughout the lifespan of affected children [6,45]. In our study, many respondents were exposed to aggressive acts (78.7%) and neglect (48.9%) from their elementary to high school periods. Because the cognitive dysfunction associated with many mental disorders impairs functionality, this may have impaired the parental performance of household chores [46], leading to neglect of their children. In Japan, physical violence toward family members occurs in 60% of schizophrenia sufferers [39]. Severe direct violence or neglect is categorized by the child protection system as abuse, but none of the respondents in this study were placed under child protection. Because they were considered insufficiently exposed to significant levels of violence or neglect, these experiences seemed to be unrelated to distress in the current study. However, these results do not mean that neglect or exposure to aggressive acts has no serious impact on children. The results merely suggest that for COPMIs, the long-term effects of emotional caregiving upon psychological distress are more significant compared with experiences of abuse or neglect.

In the current study, 73.4% of the respondents had the experience of providing emotional care to their parents. Mental illness requires families to become emotionally involved with the patients. Earlier qualitative studies have shown that children observe and react to the unpredictability of their parents’ illness-related behavior [19,47]. As a result, children carefully monitor their parents for signs of change [48] and try to manage these behaviors to maintain a safe home environment [19]. The consequences of providing long-term emotional care may continue to affect children later in life. As adults, young carers may find it difficult to establish trust with others [49]. In their own childhoods, the normal roles of parent and child in care provision were reversed as they often had to provide care for their parents [20]. Therefore, they were never able to truly live as children. Such childhood experiences have been described in previous studies carried out in other countries and are often shared during KODOMO-PEER meetings in Japan.

### 4.3. Implications for Practice

The results of this study suggest that providing emotional care for parents with mental illness has long-term effects on the psychological health of the children involved. Many of our survey respondents grew up experiencing their parents’ illness and providing emotional care for them. In Japan, although support for victims of child abuse is well established [11], support for young carers has only just begun. The findings reported here suggest a need to reduce the burden of emotional care in particular. Emotional care should not be provided by children but by visiting community nurses or other professionals on behalf of children. When health professionals support parents with childcare responsibilities, they also need to keep in mind that when children take on care roles, the burden typically falls most heavily on girls. However, whether COPMIs are caring for their parents or struggling in other ways with family matters, they are often reluctant to seek help because of stigma [50]. In Japan, more than 80% of COPMIs did not consult their schoolteachers about conditions at home [37]. When mentally ill parents are raising children, it is desirable for the treating medical facility to proactively assess the family’s situation and link them to appropriate home-visiting services. 

Finally, because many COPMIs continue to experience psychological distress in adulthood and need assistance, more support for such individuals is also needed.

### 4.4. Research Limitations and Future Research

The limitations of this study include the following: First, the sample may be unrepresentative of the general population because of its small size, the low response rate (50%), and its recruitment through a specific group. Nevertheless, this survey involved the largest sample size of adult children of parents with mental illness of any study carried out in Japan. In addition, respondents with high levels of distress may have been unable to respond due to poor health, and there may also have been bias among respondents. Respondents often find KODOMO-PEER through the internet, and this itself may reflect their awareness of concerns and difficulties originating in childhood and persisting into adulthood. Therefore, it is unclear if the respondents fairly and accurately represent all of the children of parents with mental illness. Secondly, we were unable to determine any economic circumstances in childhood that might have affected current psychological distress [6,44] because this study used variables from a previously conducted study. Thirdly, the memory of individual respondents may also serve as a source of bias. Our survey used a self-reporting questionnaire, and the accuracy of responses could not be guaranteed. Lastly, we did not explore any experiences of childhood prior to elementary school because this survey’s main purpose was to focus on school-aged children.

Future research methods need to be more broadly based and not limited to self-help groups in order to reduce subject bias. In addition, prospective longitudinal surveys and interviews should be used to ensure accurate study information.

## 5. Conclusions

In this study, we sought to clarify whether the present-day experience of psychological distress among adult COPMIs is related to their experiences of being abused and neglected and of having provided emotional care for their parents during their childhood years, adjusted for factors potentially associated with dependent and independent variables. A logistic regression analysis revealed that the experience of providing emotional care for parents during childhood was significantly related to high levels of distress in adulthood. For children of parents with mental illnesses, the effects of providing emotional care for parents during childhood may include long-term psychological distress. The findings suggest a need to reduce the burden of emotional care provided by children.

## Figures and Tables

**Table 1 healthcare-11-00214-t001:** Demographic data of adult children from the high- and low-distress groups.

		All	Low-Distress Group (K6 ≤ 4)	High-Distress Group (K6 ≥ 5)			
		n = 94	n = 29	n = 65			
		n (%)	n (%)	n (%)	*x^2^*		*p*
Adult children							
Gender	Male	15(16.0%)	4 (13.8%)	11 (16.9%)	0.147	1.000	
	Female	79(84.0%)	25 (86.2%)	54 (83.1%)			
Age (years)	20–29	35 (37.2%)	6 (20.7%)	29 (44.6%)	9.190	0.027 *	
	30–39	21 (22.3%)	7 (24.1%)	14 (21.5%)			
	40–49	16 (17.0%)	4 (13.8%)	12 (18.5%)			
	50 or older	22 (23.4%)	12 (41.4%)	10 (15.4%)			
Have siblings	Yes	54 (57.4%)	20(69.0%)	34 (52.3%)	2.276	0.131	
	No	40 (42.6%)	9 (31.0%)	31 (47.7%)			
Mental illness	Suffered	23 (24.5%)	4 (13.8%)	19 (29.2%)	2.586	0.108	
	Not suffered	71 (75.5%)	25 (86.2%)	46 (70.8%)			
Physical	Poor	39 (41.5%)	4 (13.8%)	35 (53.8%)	13.252	<0.001 *	
condition	Not poor	55 (58.5%)	25 (86.2%)	30 (46.2%)			
Separation from	Experienced	41 (43.6%)	14 (48.3%)	27 (41.5%)	0.370	0.543	
parents	Not experienced	53 (56.4%)	15 (51.7%)	38 (68.5%)			
Parents with mental illness							
Parents with	One	79 (84.0%)	25 (86.2%)	54 (83.1%)	0.146	1.000	
mental illness	Both	15 (16.0%)	4 (13.8%)	11 (16.9%)			
Treatment	Continued,	49 (52.1%)	15 (51.7%)	34 (52.3%)	0.003	0.958	
	Discontinued or never treated	45 (47.9%)	14 (48.3%)	31 (47.7%)			
Spousal	Presence	60 (63.8%)	17 (58.6%)	43 (66.2%)	0.493	0.483	
presence	Absence	34 (36.2%)	12 (41.4%)	22 (33.8%)			

*p*-values were calculated for the differences between high- and low-distress groups using the χ^2^ test. * *p* < 0.05.

**Table 2 healthcare-11-00214-t002:** Adverse experiences in childhood in the high- and low-distress groups.

		All	Low-Distress Group (K6 ≤ 4)	High-Distress Group (K6 ≥ 5)		
		n = 94	n = 29	n = 65		
		n (%)	n (%)	n (%)	*x^2^*	*p*
Exposure to aggressive acts	Elementary	66 (70.2%)	16 (55.2%)	50 (76.9%)	4.536	0.033 *
	Junior high	67 (71.3%)	17 (58.6%)	50 (76.9%)	3.281	0.070
	High	50 (53.2%)	15 (51.7%)	35 (53.9%)	0.036	0.849
	Ever	74 (78.7%)	18 (62.1%)	56 (86.2%)	6.945	0.008 *
Neglect	Elementary	31 (33.0%)	7 (24.1%)	24 (36.9%)	1.483	0.223
	Junior high	35 (37.2%)	7 (24.1%)	28 (43.1%)	3.078	0.079
	High	25 (26.6%)	4 (13.8%)	21 (32.3%)	3.521	0.061
	Ever	46 (48.9%)	8 (27.6%)	30 (46.2%)	2.871	0.090
Emotional care	Elementary	56 (59.6%)	12 (41.4%)	44 (67.7%)	5.765	0.016 *
	Junior high	57 (60.6%)	11 (37.9%)	46 (70.8%)	9.060	0.003 *
	High	57 (60.6%)	13 (44.8%)	44 (67.7%)	4.392	0.036 *
	Ever	69 (73.4%)	16 (55.2%)	53 (81.5%)	7.141	0.008 *

*p*-values were calculated for the differences between high- and low-distress groups using the χ^2^ test. * *p* < 0.05.

**Table 3 healthcare-11-00214-t003:** Risk factors for adult children with high distress (K6 ≥ 5).

		OR	95% CI	*p*
Provide emotional care	Never experienced	1.00	Reference	
	Ever experienced	3.48	1.21–9.96	0.020 *
Physical condition	Not poor	1.00	Reference	
	Poor	7.15	2.17–23.58	0.001 *

CI: confidence interval. * *p* < 0.05. Multiple logistic regression: The following variables were selected via stepwise method from exposure to aggressive acts, neglect, emotional care, gender, age, having siblings, suffering mental illness, physical condition, separation from parents, parents with mental illness (one or both), parental treatment, and the parental spousal presence.

## Data Availability

The data of this study are not publicly available to protect the participants’ privacy.

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
