# Peer review of "Childhood Adversities and Psychological Health of Adult Children of Parents with Mental Illness in Japan"

_healthcare, 2023, doi:10.3390/healthcare11020214_

Round 1

Reviewer 1 Report

The research topic is important and relatively new. Also, the researchers put a great effort collecting the data. Additionally, this research is significant to provide information and direction mental illness about psychological health among adult children in Japan. However, the researcher hasn't used profound review, methods, some findings and conclusion could benefit from the following comments:

Abstract

The abstract section is not a clear problem statement and then followed as purpose of study. For instance, the author stated “parental mental illness is considered one of many adverse childhood experiences”. I am confused, what are many adverse childhood experiences? How many factors impacts of childhood experiences? It is confusing sentences.

Moreover, the conclusion and implications of study are missing. How does social practical? How do healthcare contributions?

Introduction

- The first, second, third, and four paragraphs of introduction (line 34-77, page 1) should be paraphrased. It is not clear what the authors mean by saying what is most important about psychological health of adult children of parents with mental illness in Japan. Did they mean how the relationships between abuse, neglect, and the emotional care provided by children at school-going age of parents with mental illness in their roles as young carers, and the current psychological distress in adulthood?

1.  Should be cleared why is significant?

2.  How many factors are affected psychological health linked to mental illness?

3.  How are gaps existing in japan context?

4.  What objectives are filled in the contexts?

The objectives are purposed to test relationships between the emotional care and what? I found there are so many factors, but the author never defined clearly. For example, emotional care, including addition, abuse, and so what? I am a professor in quantitative research, but I am confused what and how many factors can be tested. Do not include all factors/variables/indicators/items in one sentence, which makes the readers confused what did you test?

Literature review

A wide and diverse literature has been used to develop the arguments presented in the paper. Given that there is a limited number of studies that have investigated the mental illness from the multidimensional approach, this paper has made a concerted effort to include the literature. However, it would have helped within including a literature review section in the paper that clearly highlights the gaps in the literature.

My suggestions would be to have a literature review sections that will then lead into the identification of propositions or factors/variables such as psychological health and mental illness of adult children between childhood and parents. It is not clear if there is a need to even consider the different psychological health effects on mental illness. As indicated in the paper, the mental illness is to some extent correlated. The following are some of the questions I think would have been addressed in a literature review.

1. Why does the research/research decide to investigate the psychological health and mental illness in totality as opposed to unidimensional?

2. What do you mean by psychological health and mental illness of childhood and parents? How does the study operationalize these concepts to allow collection of data?3. How many factors did you conceptualize from theoretical approaches to psychological health and mental illness?

Methods

The author did not clearly conduct research design through data analysis? What criteria of study selection the participants? How were the articles chosen? 

It is uncleared what and how do methods (design, setting, data collection, measures, data analysis) approach to psychological health and mental illness of childhood and parents.

Line 95-100

Line 101-112

Line 115-124

Line 126-140 (I found is too short sentence to explain about independent variables).

Line 142-145 (control variables). I am so confused; did you control mental illness and physical conditions. Why? How? Most control variables are only nominal scale. If yes, your work is missing and wrong testing.

Results

Respondents’ demographics (line 166-168) is too short and not clear what?

3.2. Psychological distress (line 189-191) is too short and not clear what value means?

Missing control variables in the findings section.

3.4. Comparisons between the high-distress and low-distress groups (line 201-204) is too short in explaining.

Discussion

Lacks discussion what the main findings are? What is new contributed from this study? How about the other found difference/similarity? What this study contributed to fill the gaps of knowledge and practical implications? Should be discussed and clear what/how?

See line 229-245

4.3. Implications for practice

It is not why did you not provide practical and care contribution in Japan contexts.

Anyway, the writing really needs improvements and actually avoid a better understanding of the article. Furthermore, the objectives of the article are not described. Nor are contributions reported. What's new in the article? What did the presented findings, improve in relation to others (similar) scholars - what are the references that led the authors to exist in Japan context?

Reviewer 2 Report

This is a study on psychological health in adult children of parents with mental illness. Children of parents with mental illness are a sensitive and valuable population, who are vulnerable and can drag sequelae through their life span. So, this is a relevant paper.

However, I suggest the following modifications to further strengthen the manuscript:

Abstract:

1.    I suggest including age and percentage of participants of the sample.

2.    Please rewrite the sentence “A web-based cross-sectional study was conducted among 240 participants who had joined self-help groups”. It is not precise because not all the 240 subjects actually participated in this study. As you say later, although 240 were invited to participate, only 120 responded and 94 were included in the study.

Introduction

In the introduction, you defined young carers as children and young persons under 18 who provide or intend to provide care, assistance or support to another family member. Since participants of your study have over 20 years, could you add some explanation about adult children of parent with mental illness? Or about how being a child under 18 years of a parent with mental illness can affect the own adult life?

Method

Participants

Please specify the inclusion and exclusion criteria to participate in this study.

Procedure

Could you please add a Procedure section?

Ethical approval

Please include a statement about the informed consent.

Discussion

1.    Begin the discussion with a reminder of the aim of the study.

2.    Lines 242-245. Could you provide some tentative explanation for the obtained result that 24.5% of the respondents suffered from a mental illness?

3.    Line 256. According to the journal style, Kageyama et al. citation should be in number format.

4.    Research limitations: please include the response rate about 50% (120 respondents of 240 invited to participate) as a limitation of the study.

Reviewer 3 Report

This article is an extension of previous (western) findings about Adverse Childhood Experiences to young Japanese caregivers. Although some similarities to western findings are reported here, but there are others that are not clearly replicated here. The authors make surprisingly little of these similarities and differences and do little to explain them. Are there differences between Japanese and western expectations about parenting? Are there differences in stoicism, for example, which might, relative to western samples, underestimate the impact of ACEs?

It would be nice to have more comparison groups than just high and low distress (e.g., non-caregivers or caregivers for parents with physical illness). I don't know the K6, but a cut-off of 4/5 of 24 seems like a very low cut-off. It might be helpful to bring in comparison data from the Furukawa et al. (2008) validation study, if not from other samples. Nice points about your sample possibly being a nonrepresentative one.

Line 52: I'm not sure what the authors meant here, maybe "We focused on parental mental illness in the current study"???

Line 72: Substitute "linked" for "led": "The emotional care can be led to psychological distress for carers"

Lines 72-74: Combine these two sentences: "The emotional care can be led to psychological distress for carers because living with a person with severe mental illness is not easy. The changes in the behaviour and personality of in a patient can be very distressing for close family members [19–23].

Line 103: Substitute in for on: "on October and November 2019."

What does physical condition worse/not worse mean?

Lines 133-134: Add commas around the clause, as indicated here: "one or both of the following items, ‘An adequate amount of food was not provided’ and ‘The laundry and cleaning were not well done,’ were classified under ‘neglect’.

Lines 142-143: Add "and": "gender, age, having siblings, suffering mental illness, and physical condition."

Line 166: Add "the": "the majority"

Line 167: Substitute "their" for "the": "As for their current state of health,"

Indicate how old children are in each of these school age groups. This is not consistent throughout the world.

Describe your findings of your regression further. I'm not following.

Lines 230-231: "As a result, many of the respondents grew up experiencing experienced their parents’ illness throughout their lives."

Lines 249-251: "This contrasts with studies on ACEs that reported there were significant relationships between child abuse and neglect with an increased risk for psychological health problems across the child’s lifespan [6,43].

You might talk about cultural differences between this Japanese sample and the western samples that most of the research you review has been done on. You have interesting findings, but don't do as much with them as you could.

Reviewer 4 Report

please see the file attached.

Round 2

Reviewer 1 Report

This is a fine revision of the work. The paper certainly does contribute enough to warrant publication. Best of luck with the acceptance.

Author Response

We learned a lot from your useful comments. Thank you very much.